# Effect of scanning-aid agents on the scanning accuracy in specially designed metallic models: A laboratory study

Hyun-Su Oh[1], Young-Jun Lim[1]*, Bongju Kim[2]*, Myung-Joo Kim[1], Ho-Beom Kwon[1], Yeon-Wha Baek[3]

1 Department of Prosthodontics and Dental Research Institute, School of Dentistry, Seoul National University, Seoul, Republic of Korea, 2 Dental Life Science Research Institute, Seoul National University Dental Hospital, Seoul, Republic of Korea, 3 Department of Prosthodontics, Seoul National University Gwanak Dental Hospital, School of Dentistry, Seoul National University, Seoul, Republic of Korea

* limdds@snu.ac.kr (Y-JL); bjkim016@snu.ac.kr (BK)

## Abstract

The advent of intraoral scanning methods has caused a paradigm shift in dentistry. However, despite their many advantages, intraoral scanners cannot accurately recognize the metallic surfaces of prothesis. Therefore, this experiment was designed to verify the effect of scanning-aid agents on the scanning accuracy using metallic reference models. Three different types of metallic reference models (inlay, onlay, and bridge) were specially designed and produced using a milling machine to simulate intraoral dental restorations. Three experimental groups (application of ScanCure, IP Scan Spray, and VITA Powder Scan Spray) were set up and scanned images (each n = 5) were acquired using the I500® intraoral scanner. The reference datasets were established by a 3D design that reflected the deviations between the measured distances and previously planned distances on the reference models. All acquired experimental datasets were digitally superimposed and compared with the reference datasets. Intragroup comparisons (precision, n = 10) were also performed. The root mean square (RMS) values of trueness in the ScanCure and IP groups were significantly more accurate than those of the VITA group in the inlay and onlay reference models (p < 0.05). Notably, in the bridge reference model, the liquid-type ScanCure group showed the highest accuracy of trueness, with statistical significance (p < 0.05). However, the RMS values of precision were not significantly different among the groups. These findings suggest that liquid-type scanning agents can be effectively used to obtain more accurate scan images of intraoral metallic dental restorations.

## Introduction

The development of intraoral scanning technology has caused a paradigm shift from conventional impression techniques to direct digital impression making [1,2]. With computer-aided design (CAD) / computer-aided manufacturing (CAM) systems and clinically reliable intraoral scanners, practitioners can easily obtain intraoral scanned images to fabricate the

**Data Availability Statement:** All relevant data are within the manuscript.

**Funding:** This work was supported by the Korea Medical Device Development Fund grant funded by

the Korea government (the Ministry of Science and ICT, the Ministry of Trade, Industry and Energy, the Ministry of Health & Welfare, the Ministry of Food and Drug Safety) (Project Number: 1711138936, RS-2020-KD000291). The funders had no role in study design, data collection and analysis, decision to publish, or preparation of the manuscript. There was no additional external funding received for this study.

**Competing interests:** The authors have declared that no competing interests exist.

prosthesis digitally [3]. Additionally, as the accuracy of conventional impressions affects the final fit of the dental prosthesis, precise digital scanned images are a prerequisite for making clinically acceptable prostheses [4]. Although intraoral scanning methods have several advantages such as saving storage space, patient convenience, and freedom from distortion errors in impression material, there are also crucial limitations in digital image acquisition in intraoral environments [5,6].

First, intraoral scanners used in the dental field cannot accurately recognize the metallic surfaces of the prostheses. Because scanners obtain digital information by interpreting the diffusion of light from the surface of objects, the shiny or translucent properties of the metallic surface interfere with the matching of the point of interest by the software due to overexposure [7,8]. This results in incorrectly scanned images and consequently, in an improperly fitted prosthesis. Therefore, to create a homogeneous antireflective surface, titanium dioxide powder is often applied before intraoral scanning [9]. Of course, this additional step can be uncomfortable for both patients and clinicians [10]. The thickness of the scanning powder applied to the objects can vary and is affected by the proficiency of the operator [11]. However, the software of the intraoral scanner itself can compensate for these differences [12]. Liquid-type coating agents can achieve a more uniform coating thickness, and therefore, a more accurate scanned image, than the powder-type agents [13].

Second, errors from the increased scan distance are inevitable with intraoral scanners [14]. The stitching or matching process is necessary to form the whole scanned image because the field of view of the intraoral scanner is too small to obtain the total image at once, especially of the full dental arch [15]. Thus, stitching errors are accumulated in the software proportional to the scan distance [16]. These also results in an inappropriate final prosthesis. To minimize these scanning errors over a long span, such as a full dental arch, several factors are considered. For instance, not only the matching algorithms of software, but also the sensor quality of the intraoral scanner must be improved [17]. Scanning strategies such as scanning path, environment control, and application of scanning-aid agents are used to obtain accurate scanned images [10,18].

As mentioned above, reflective metallic surfaces are challenging to scan using intraoral scanners. In addition, the accuracy of the scanned image may be affected by the extent of the scan. Therefore, to reduce the reflective properties of the metallic surface and increase the accuracy in the case of a long span, surface treatments, such as the application of scanning-aid agents, are frequently suggested. A previous study reported that when a resin-based full-arch model was scanned with the application of scanning-aid materials, the precision of root mean square (RMS) values was improved [18].

The purpose of this study was to evaluate the influence of scanning-aid materials on the scanning accuracy in specially designed metallic models that imitate intraoral dental restorations, such as inlay, onlay, and bridge. To compare the scanning accuracy, 3D superimposition with the best-fit algorithm was used.

## Materials and methods

### Reference models and scanning-aid materials

Three different types of metallic reference models were designed using CAD software (Solidworks 2016, Dassault Systèmes SolidWorks Corp., Waltham, MA, USA) and milled using a 5-axis milling machine (VARIAXIS i-600, Yamazaki mazak Corp., Aichi, Japan) (Fig 1). The metallic material used for milling the reference model was an aluminum alloy. These reference models simulated different crown shapes for dental restorations: inlay, onlay, and bridge. The

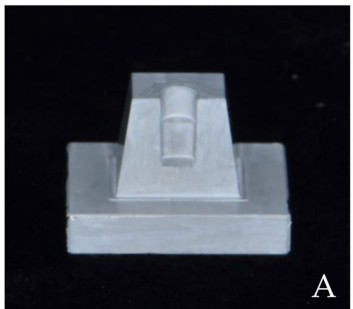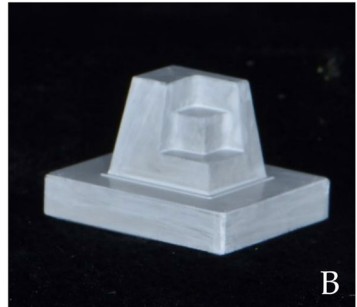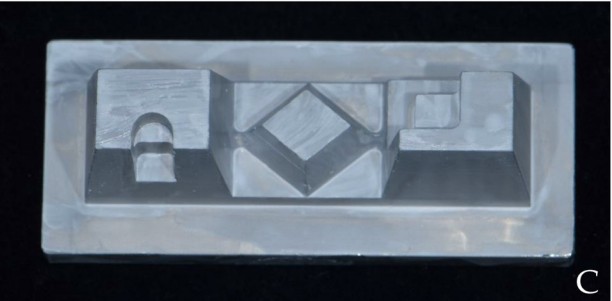

**Fig 1. Specially designed reference models.** A—inlay model; B—onlay model; C—bridge model.

bridge model was designed by connecting one inlay and one onlay form with one crown form (Fig 1C).

Three different types of scanning-aid agents were used: Vita Powder Scan Spray (Vita Zahnfabrik, Germany), IP Scan spray (IP-Division, Haimhausen, Germany), and ScanCure (SC-80, ODS Co., Incheon, Korea). The VITA and IP spray were powder-type agents, while ScanCure was liquid. VITA Powder Scan Spray is a spray-on, titanium dioxide free and blue colored pigment suspension with ethanol and isobutane. IP Scan Spray contains titanium dioxide with ethanol, propane, butane, and isobutane. And the ScanCure contains titanium dioxide and ethanol.

Because the application procedure of scanning-aid materials is sensitive to operator's proficiency, one skilled prosthodontist applied the materials on the model as follows: the liquid type of ScanCure was applied by one brush stroke at a time in all surfaces of model, powder type of VITA and IP scan spray was applied at a same distance (5 inch) and angulation (45 degree) of the spray tip from the specimen with the same time (2 seconds) to make an uniform thin layer of powder.

## Acquisition of digital data

Because it was difficult to acquire the scanned images of the untreated metallic reference models (Fig 2), the reference datasets were obtained using a 3D design, which reflected the errors between the measured distances and the previously planned distances on the reference models. The distances on the fabricated reference models were calculated using a laser scanner (HERO7106, Dukin Corp.,Daejeon, Korea).

The experimental datasets of the three groups (application of Vita Powder Scan Spray (VITA), IP Scan spray(IP), and ScanCure) were obtained using the I500 intraoral scanner (Medit Co., Seoul, Korea) with the different reference models (inlay, onlay, and bridge). Five scan images were obtained for each experimental group, and all scanned datasets were converted into the standard tessellation language (STL) data format.

The scanning procedure was executed by a skilled prosthodontist. The remnants of the scanning-aid materials after scanning were removed using organic solvents and an air compressor water gun. The scanning-aid agents were then reapplied for each scanning cycle.

## Three-dimensional analysis

All experimental STL datasets were compared with the reference dataset (trueness, n = 5) using the best-fit superimposition method of the 3D analysis software (Geomagic Control X®, 3d systems, Rock Hill, USA). Intragroup comparisons were performed for each experimental group (precision, n = 10). The alignment setting value was determined to produce

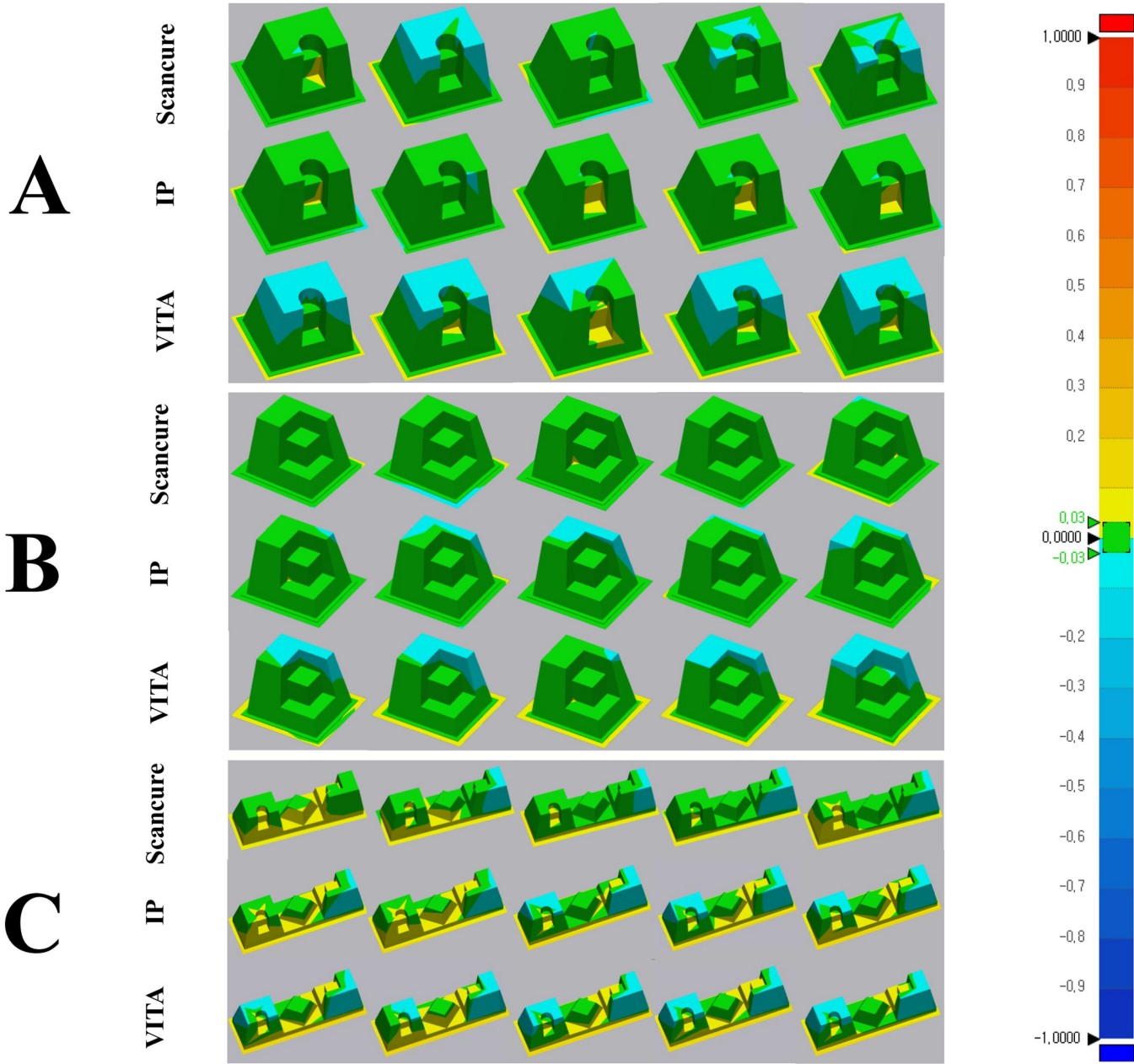

**Fig 2. Scanned images with a model scanner (Identica T500®).** A—inlay model; B—onlay model; C—bridge model.

minimal error based on least square regression with a set tolerance of ± 0.03 mm and a maximum tolerance range of ± 0.3 mm. The mean and standard deviation of the experimental groups were measured using the root mean square (RMS) value.

## Statistical analysis

Statistical analysis was performed using the SigmaPlot 14.0™ (Systat Software Inc., San Jose, CA, USA) program. Differences between groups in trueness, precision were evaluated by Kruskal–Wallis test. Pairwise comparisons were performed through the Mann-Whitney test in

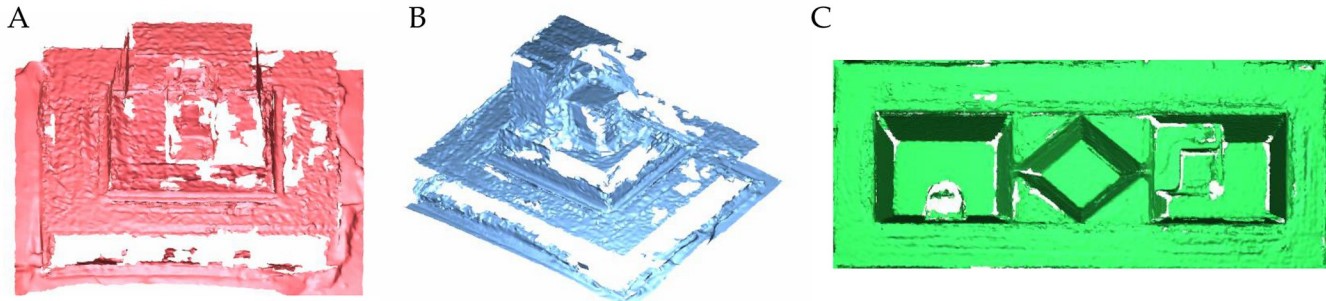

**Fig 3. 3D superimposition color maps of three experimental groups in three reference models.** A—inlay model; B—onlay model; C—bridge model (tolerance range ± 30 μm).

the case of significant difference according to the Kruskal–Wallis test. The level of significance ($\alpha = 0.05$) was adjusted according to the Bonferroni correction method.

## Results

### Trueness

In each reference model, five datasets (trueness, n = 5) for each group (ScanCure, IP, and VITA) were superimposed with the reference datasets in the 3D analysis software (Fig 3). The mean and standard deviation (SD) of the RMS values for each group were measured. The results are summarized in Table 1.

In the inlay and onlay reference models, the RMS values of the ScanCure and IP groups were significantly lower than those of the VITA group ($p < 0.05$) (Fig 4A and 4B). In the bridge reference model, the ScanCure group had the lowest RMS value compared to the other groups, and the IP group had a significantly lower RMS value than the VITA group ($p < 0.05$) (Fig 4C).

### Precision

Similarly, the RMS values (precision, n = 10) for each group were measured using a combination of the two different intragroup datasets. The mean and standard deviation of the RMS values are shown in Table 2.

Except for the RMS values of trueness, there were no statistically significant differences among the three experimental groups with all types of reference models ($p > 0.05$) (Fig 5).

## Discussion

In terms of trueness, the ScanCure and IP groups had significantly lower RMS values than the VITA group with the inlay and onlay reference models; with the bridge model, the RMS value of the ScanCure group was the lowest among the three groups with a statistically significant difference. These results may be explained by the properties of each scanning-aid material.

**Table 1. The RMS values of three experimental groups (ScanCure, IP, VITA) in each reference model (mean ± SD).**

| Trueness | n | ScanCure (μm) | IP (μm) | VITA (μm) |
|---|---|---|---|---|
| Inlay | 5 | 46.06 ± 2.56 | 45.46 ± 1.80 | 62.74 ± 2.73 |
| Onlay | 5 | 45.34 ± 3.94 | 47.10 ± 2.82 | 69.10 ± 4.15 |
| Bridge | 5 | 64.62 ± 4.27 | 84.38 ± 1.80 | 98.94 ± 1.11 |

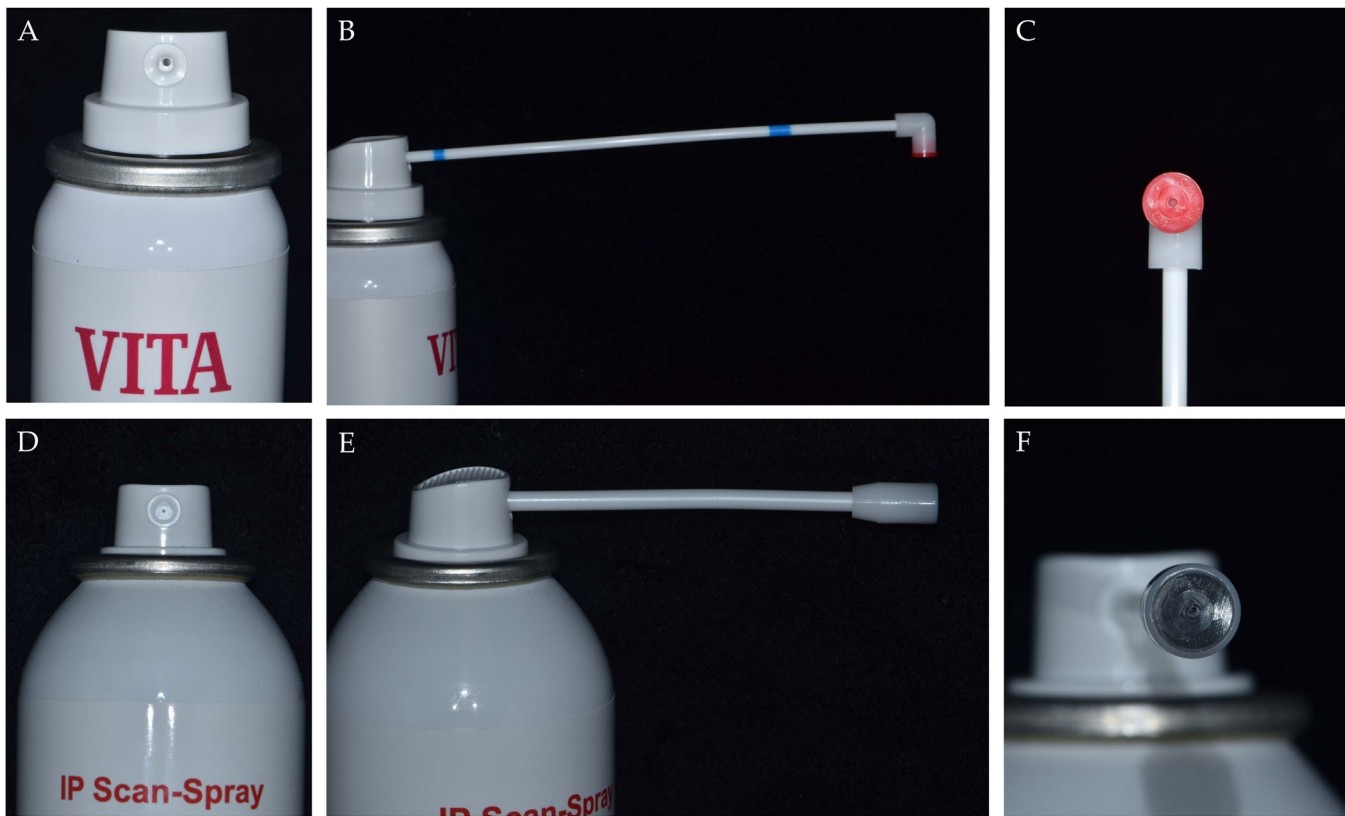

**Fig 4. Box plots of the RMS values for three experimental groups (Scancure, IP, VITA) in three reference models.** A–inlay; B–onlay; C—bridge.

Unlike the powder-spray types (IP, VITA), ScanCure is a liquid-paint type agent. Thus, ScanCure can be applied to hard-to-reach surfaces of the model as well [13]. In addition, the properties of surface tension, contact angle, and viscosity of the liquid components help maintain a uniform thickness on the surface [19]. In the ScanCure group, the uniformity of the coating layer was confirmed by sight; the layer was also maintained well on the metallic reference models (Fig 6).

Furthermore, the VITA group had the largest RMS values among the three groups with all the reference models, with a statistically significant difference. When using the powder-type agents, the total amount that is sprayed on the models may vary even when the same operator applies them under the same conditions (same angle, distance, time, etc.). This is because the design and size of the injection nozzles are different for each agent (Fig 7). The VITA agent had a larger ejection hole size than the IP agent (Fig 7C and 7F). This difference led to a larger amount of the VITA agent being applied; this explains the greater scanning errors seen in the VITA group compared to the IP group. In addition, the powder-type agents were more

**Table 2. The RMS values of three experimental groups (ScanCure, IP, VITA) in each reference model (mean ± SD).**

| Trueness | n | ScanCure (μm) | IP (μm) | VITA (μm) |
|---|---|---|---|---|
| Inlay | 10 | 35.21 ± 8.39 | 31.50 ± 14.26 | 31.98 ± 6.91 |
| Onlay | 10 | 35.07 ± 8.51 | 30.26 ± 8.82 | 35.49 ± 8.15 |
| Bridge | 10 | 50.68 ± 8.49 | 49.34 ± 12.23 | 58.75 ± 12.66 |

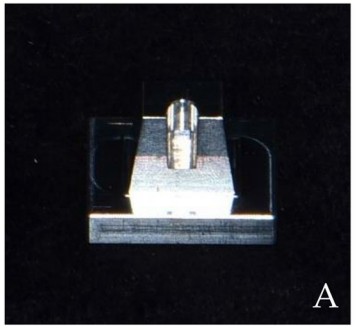
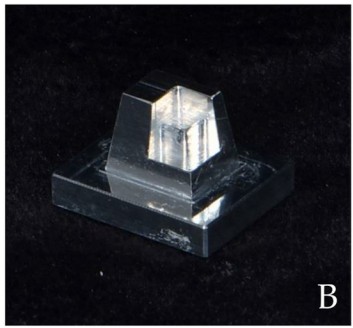
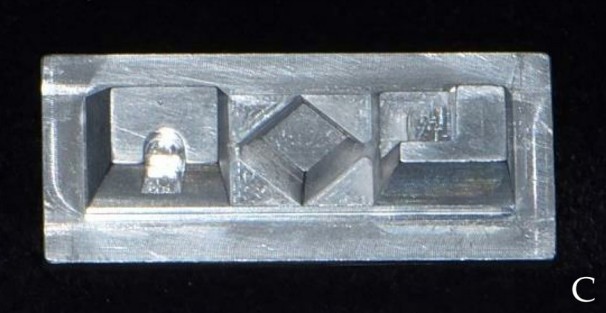

**Fig 5. Box plots of the RMS values for three experimental groups (Scancure, IP, VITA) in three reference models.** A—inlay model; B—onlay model; C—bridge model.

susceptible to other external conditions (e.g., the size of the object to be applied, operator, nozzle design, spraying time, distance, and angle) than the liquid type.

Precision is defined as the reproducibility of a scanned image under the same conditions [20]. Unlike trueness, there were no statistically significant differences in the RMS values of precision even though the IP group showed the highest precision among the three groups with all the reference models. As described above, the trueness values were significantly different among the groups because the total amount and uniformity of the applied materials were different. However, in each group, the hardware design of the scanning-aid materials was sufficient to reproduce the results of the scanned images. The reproducibility and accuracy of the outcomes were affected by the operator [21]. This means that highly skilled prosthodontists can reduce the errors in intragroup comparisons. In addition, in all three experimental groups, the RMS values of the bridge reference model were larger than those of the inlay and onlay reference models. This suggested that the error accumulates as the scan area increases because the intraoral scanner acquired the digital image of the object using the stitching algorithm [16].

The limitation of this study was that it did not reflect real clinical situations (e.g., accessibility of scanner in the mouth, patient's compliance, humidity environment such as saliva, blood, etc.). Moreover, it is suggested that other types of intraoral scanners such as confocal-type (e.g., 3Shape Trios®) and not the triangulation-type (Medit I500®) should be used. Further

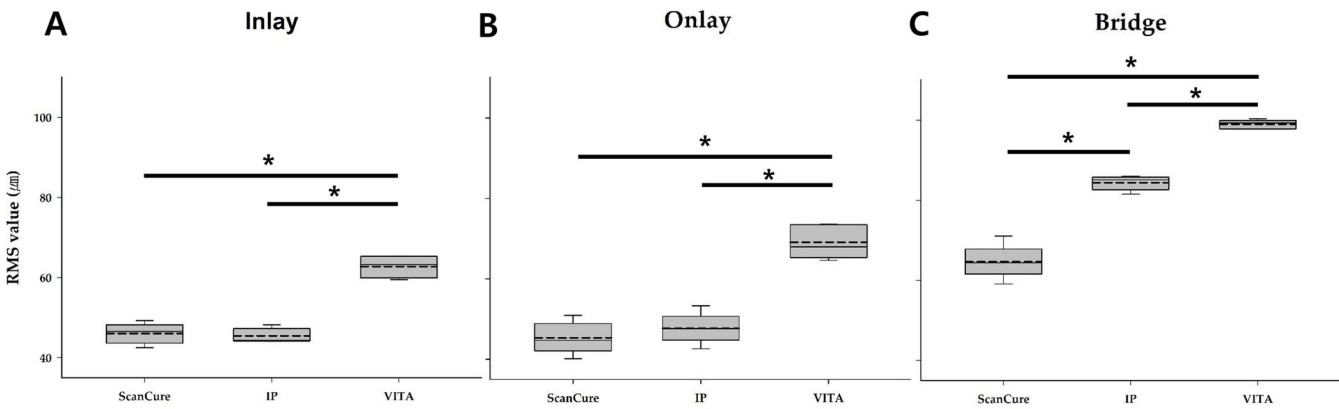

**Fig 6. Application of ScanCure® agent on the three reference models.** A—inlay model; B—onlay model; C—bridge model.

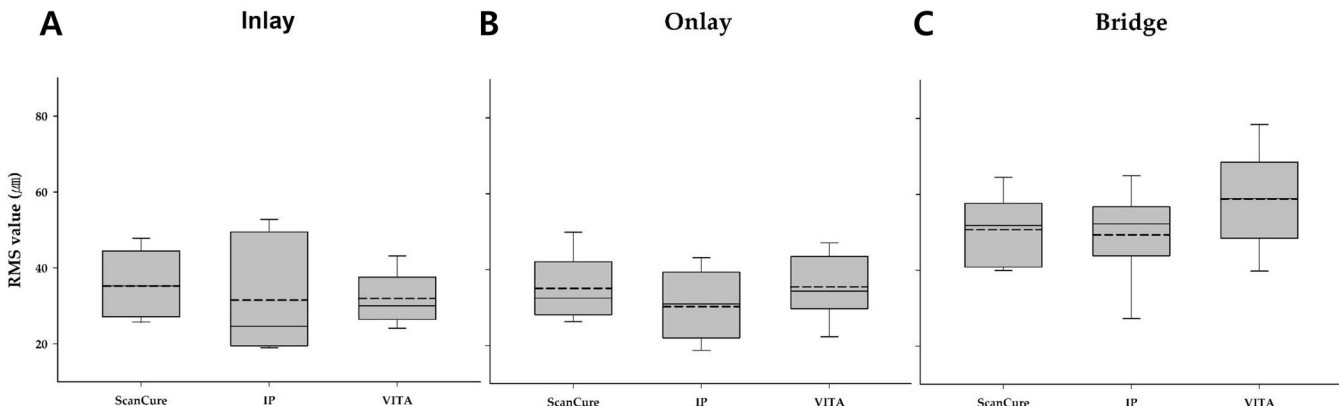

**Fig 7. Designs of ejection nozzle and spraying hole in powder-type agents.** A—without ejection nozzle in VITA; B—with ejection nozzle in VITA; C—spraying hole in VITA; D—without ejection nozzle in IP; E—with ejection nozzle in IP; F—spraying hole in IP.

studies using the full-arch reference model are necessary to verify the effect of the scan distance on the scanning accuracy.

## Conclusions

The liquid-type scanning-aid agent showed a significantly lower RMS value of trueness than the other powder-type agents with the metallic bride reference model. This suggests that liquid-type scanning-aid agents can be applied uniformly on the entire prosthetic surface by the brushing technique, especially when large areas need to be scanned. In addition, it is relatively difficult to control the amount of powder-type agent applied. (Separate the paragraphs)

Based on our findings, we recommend that liquid-type scanning agents should be used to obtain more accurate scan images of the metallic surfaces of dental restorations in clinical practice.

## Author Contributions

**Conceptualization:** Hyun-Su Oh, Young-Jun Lim.

**Data curation:** Hyun-Su Oh, Bongju Kim, Myung-Joo Kim.

**Formal analysis:** Hyun-Su Oh, Bongju Kim.

**Investigation:** Hyun-Su Oh, Young-Jun Lim, Bongju Kim, Myung-Joo Kim, Ho-Beom Kwon, Yeon-Wha Baek.

**Methodology:** Young-Jun Lim, Bongju Kim.

**Project administration:** Young-Jun Lim.

**Software:** Hyun-Su Oh.

**Visualization:** Bongju Kim, Ho-Beom Kwon, Yeon-Wha Baek.

**Writing – original draft:** Hyun-Su Oh, Young-Jun Lim.

**Writing – review & editing:** Hyun-Su Oh, Young-Jun Lim, Bongju Kim.

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
