## [Decision Letter · Decision Letter 0]

23 Feb 2022

PONE-D-21-40935Effect of Scanning-Aid Agents on the Scanning Accuracy in Specially Designed Metallic Models: A Laboratory StudyPLOS ONE

Dear Dr. Lim,

Thank you for submitting your manuscript to PLOS ONE. After careful consideration, we feel that it has merit but does not fully meet PLOS ONE’s publication criteria as it currently stands. Therefore, we invite you to submit a revised version of the manuscript that addresses the points raised during the review process.

We look forward to receiving your revised manuscript.

Kind regards,

Antonio Riveiro Rodríguez, PhD

Academic Editor

PLOS ONE

Journal Requirements:

(This work was supported by the Korea Medical Device Development Fund grant funded by the Korea government (the Ministry of Science and ICT, the Ministry of Trade, Industry and Energy, the Ministry of Health & Welfare, the Ministry of Food and Drug Safety) (Project Number: 1711138936, KMDF_PR_20200901_0291).)

(The funders had no role in study design, data collection and analysis, decision to publish, or preparation of the manuscript.)

(This work was supported by the Korea Medical Device Development Fund grant funded by the Korea government (the Ministry of Science and ICT, the Ministry of Trade, Industry and Energy, the Ministry of Health & Welfare, the Ministry of Food and Drug Safety) (Project Number: 1711138936, KMDF_PR_20200901_0291).)

Reviewers' comments:

Reviewer's Responses to Questions

**Comments to the Author**

1. Is the manuscript technically sound, and do the data support the conclusions?

Reviewer #1: Partly

Reviewer #2: Yes

Reviewer #3: Yes

2. Has the statistical analysis been performed appropriately and rigorously? 

Reviewer #1: Yes

Reviewer #2: Yes

Reviewer #3: Yes

3. Have the authors made all data underlying the findings in their manuscript fully available?

Reviewer #1: Yes

Reviewer #2: Yes

Reviewer #3: Yes

4. Is the manuscript presented in an intelligible fashion and written in standard English?

Reviewer #1: Yes

Reviewer #2: Yes

Reviewer #3: Yes

5. Review Comments to the Author

Reviewer #1: The authors submit an article "Effect of Scanning-Aid Agents on the Scanning Accuracy in Specially Designed

Metallic Models: A Laboratory Study".

The study compares three types of sprays that are applied to the metal surface of dental prostheses and subsequently scanned with an intraoral scanner. Is there often a need in dental practice to scan metal surfaces of prostheses with an intraoral scanner? What other significant difference is there used between sprays (other than liquid type and powder type)?

The work scans the metal surface of prostheses - wouldn't it be enough to sandblast the surface to eliminate the shiny surface?

Cobalt chromium alloy or titanium alloy, is usually used in dental practice. Why did the authors use an aluminum alloy? It has no surface properties other than cobalt chromium alloy or titanium alloy?

In the article, you state that you used the I500 intraoral scanner for scanning - what technology does it scan with? Optical light? Laser? What is the accuracy and speed of scanning? Have you taken into account Trueness and Precision of the intraoral scanner itself when evaluating the data obtained?

The article is clearly written, described in detail Materials and Methods. The Trueness and Precision data are evaluated in the Results chapter.

In Conclusion, it is stated that it is better to use a liquid type spray.

This is certainly important information for dental practice, but I still miss the point, the main idea of the article. Is it a comparison of sprays or a comparison of metallic reference models (inlay, onlay, and bridge)? Why is it important for the dental technician / dentist which spray is better?

Reviewer #2: The report is well written and easy to understand. Accurate results have been achieved. The information is also accurately illustrated. The authors work with modern technologies in design. Although the SolidWorks software is not the latest version (version 2016 is listed), it is strong enough functionally.

Not as a remark, but as a recommendation:

- If possible, use institutional contact emails.

- The conclusion is too small. It should increase the size. (Also in conclusion) The most important points should be noted in separate paragraphs.

Reviewer #3: This is a fascinating study about the effect of scanning-aid agents on scanning accuracy in specially designed metallic models. Hence, the study adds to the current knowledge in this relatively new field and is definitely of clinical significance. However, it is still not clear to me the following matters as far as I read the manuscript.

This study aimed to evaluate the effect of different scanning-aid agents on the scanning performance of intraoral scanners. However, the description of ScanCure in the test group is insufficient. It is a liquid, so the authors need to describe more about it, including the ingredients. Also, there should be an explanation of how the agent from each group is applied. In what order and how many times these were applied or sprayed on the model. It is also necessary to describe how much powder was sprayed each time. The distance and angulation of the spray tip from the specimen are essential.

In the 'Acquisition of Digital Data' section, the authors say that it was difficult to obtain a reference dataset, so it was obtained using 3D design, but it is not easy to understand. First of all, HERO7106 equipment is a contact scanner known as a Coordinate Measuring Machine (CMM), not a laser scanner. As I understand it, to scan with a laser scanner, the spray must be applied no matter how industrial-level accurate scanners are, so the quality of scan data is no different from that of the test group intraoral scanner. By reflecting the dimensions of each block measured by CMM in the design data (mentioned as 3D design in the manuscript), the data was transformed into the same size as the real milled. And then, this modified data was used as reference data. I want to check whether I understood well. In addition, no matter how accurate measured values are reflected as design data, it is thought that there might still be errors. How did the authors try to overcome this part?

Tables 1 and 2 do not contain information on statistical results. In the discussion section, the authors mentioned trueness, but more analysis on precision results should follow. Although there was no statistical difference, the ScanCure group showed high precision, which should be added to the manuscript. The first sentence of the second paragraph in the discussion section says '~ VITA group had lowest RMS values,' but it seems to be the opposite. In the middle of the third paragraph, 'A hardware design of the products was sufficient to reproduce the results,' it is unclear whether the products refer to sprays/liquid, intraoral scanner, or specimen, so it is difficult to understand the intention of the authors.

6. PLOS authors have the option to publish the peer review history of their article (what does this mean?). If published, this will include your full peer review and any attached files.

Reviewer #1: No

Reviewer #2: **Yes: **Professor (Associate) Tihomir Dovramadjiev PhD Eng.

Reviewer #3: No

---

## [Author Response · Author response to Decision Letter 0]

29 Mar 2022

Dear editor

Manuscript Number: PONE-D-21-40935

“Effect of Scanning-Aid Agents on the Scanning Accuracy in Specially Designed Metallic Models: A Laboratory Study" 

We thank the reviewers for their comments, and we are grateful for the opportunity to provide further revisions to our paper. We changed our manuscript according to the reviewers’ comments and recommendations. We are trying to adequately address each of the points made by the reviewers. We would be very thankful if you could please reconsider a thoroughly revised manuscript. We highlighted the changes made in the manuscript by using a different color font (red): see correction marked form file, and explained details in this letter.

Response to Reviewer #1 Comments

The authors submit an article "Effect of Scanning-Aid Agents on the Scanning Accuracy in Specially Designed Metallic Models: A Laboratory Study". The study compares three types of sprays that are applied to the metal surface of dental prostheses and subsequently scanned with an intraoral scanner. Is there often a need in dental practice to scan metal surfaces of prostheses with an intraoral scanner? What other significant difference is there used between sprays (other than liquid type and powder type)?

Response: Thank you for your valuable comment. Scanning-aid materials used in this study are often necessary in daily clinical practice because the metallic shiny surface of prostheses(ex; metal crown or restoration) prevent the intraoral scanner from recognizing and obtaining the surface images properly. Composition and color are different between scanning-aid materials. According to each product information, ScanCure and IP Scan Spray contain titanium dioxide as their main components and have a white color, whereas VITA Powder Scan Spray is a spray-on, titanium dioxide free and blue colored pigment suspension. 

The work scans the metal surface of prostheses - wouldn't it be enough to sandblast the surface to eliminate the shiny surface?

Response: Thank you for your valuable comment. The dental sandblasting is usually used to increase bonding strength during the cementation of prosthesis by mechanically roughening the surface. However, if it is used to eliminate the shiny surface of prosthesis in patient’s mouth, it adversely affects the polished surface of prosthesis which is essential for oral hygiene care. Also, the powder in sandblasting rather than the specialized scanning-aid agents can cause more discomforts for patients. 

3. Cobalt chromium alloy or titanium alloy, is usually used in dental practice. Why did the authors use an aluminum alloy? It has no surface properties other than cobalt chromium alloy or titanium alloy?

Response: Thank you for your valuable comment. Due to the limitation of material selection in making the specially designed metallic models, we are focused on simulating the shiny surface of metallic model which interferes the scanner’s recognition of scanned images

As you mentioned, further studies with cobalt chromium alloy or titanium alloy in real patient’s mouth are necessary to reflect the real clinical situation.

4. In the article, you state that you used the I500 intraoral scanner for scanning - what technology does it scan with? Optical light? Laser? What is the accuracy and speed of scanning? Have you taken into account Trueness and Precision of the intraoral scanner itself when evaluating the data obtained?

Response: Thank you for your valuable comment. The I500 intraoral scanner uses video-type scanning based on triangulation technology with optical light. The accuracy is described by two measurement methods: trueness and precision. Trueness refers to the deviation of the measured value from the original value whereas precision refers to the closeness between repeated measured value. The speed of scanning was evaluated by measuring the time to obtain the total scanned image. We have considered the trueness and precision of the intraoral scanner by the product information which is based on the experiments from their own company and several published journals. Further studies using other types of intraoral scanners such as confocal-type type (e.g., 3Shape Trios®) are necessary to verify the accuracy between intraoral scanners.

5. The article is clearly written, described in detail Materials and Methods. The Trueness and Precision data are evaluated in the Results chapter. In Conclusion, it is stated that it is better to use a liquid type spray. This is certainly important information for dental practice, but I still miss the point, the main idea of the article. Is it a comparison of sprays or a comparison of metallic reference models (inlay, onlay, and bridge)? Why is it important for the dental technician / dentist which spray is better?

Response: Thank you for your valuable comment. The main point of this article is that how different types of scanning-aid materials affect the scanning accuracy using an intraoral scanner in specially designed metallic models. Inlay, onlay and bridge models were individually evaluated because we wanted to know if the same results of accuracy of scanning-aid materials were obtained in the cases of different shapes and distances. Particularly, in the bridge model, there were significantly differences of the trueness among the three different scanning-aid materials. It is important for the dental staff to know which spray is better because the accuracy of scanned image affects the success of the long-term prognosis of prosthesis 

 

Response to Reviewer #2 Comments

The report is well written and easy to understand. Accurate results have been achieved. The information is also accurately illustrated. The authors work with modern technologies in design. Although the SolidWorks software is not the latest version (version 2016 is listed), it is strong enough functionally.

Response: Thank you for your favorable review for our manuscript.

Not as a remark, but as a recommendation:

- If possible, use institutional contact emails.

Response: Thank you for your valuable comment. “limdds@snu.ac.kr and bjkim016@snu.ac.kr” is the email of the Seoul National University. 

- The conclusion is too small. It should increase the size. (Also in conclusion) The most important points should be noted in separate paragraphs.

Response: Thank you for your valuable comment. The following sentence was separated from the original paragraph as suggested by the reviewer.

Line 248: “Based on our findings, we recommend that liquid-type scanning agents should be used to obtain more accurate scan images of the metallic surfaces of dental restorations in clinical practice.”

 

Response to Reviewer #3 Comments

This is a fascinating study about the effect of scanning-aid agents on scanning accuracy in specially designed metallic models. Hence, the study adds to the current knowledge in this relatively new field and is definitely of clinical significance. However, it is still not clear to me the following matters as far as I read the manuscript.

This study aimed to evaluate the effect of different scanning-aid agents on the scanning performance of intraoral scanners. However, the description of ScanCure in the test group is insufficient. It is a liquid, so the authors need to describe more about it, including the ingredients. Also, there should be an explanation of how the agent from each group is applied. In what order and how many times these were applied or sprayed on the model. It is also necessary to describe how much powder was sprayed each time. The distance and angulation of the spray tip from the specimen are essential.

Response: Thank you for your valuable comment. The following paragraphs were added in materials and methods section as suggested by the reviewer.

Line 106: “VITA Powder Scan Spray is a spray-on, titanium dioxide free and blue colored pigment suspension with ethanol and isobutane. IP Scan Spray contains titanium dioxide with ethanol, propane, butane, and isobutane. And, ScanCure contains titanium dioxide and ethanol.”

Line 110: “Because the application procedure of scanning-aid materials is sensitive to operator’s proficiency, one skilled prosthodontist applied the materials on the model as follows: the liquid type of ScanCure was applied by one brush stroke at a time in all surfaces of model, powder type of VITA and IP scan spray was applied at a same distance (5 inch) and angulation (45 degree) of the spray tip from the specimen with the same time (2 seconds) to make an uniform thin layer of powder.”

In the 'Acquisition of Digital Data' section, the authors say that it was difficult to obtain a reference dataset, so it was obtained using 3D design, but it is not easy to understand. First of all, HERO7106 equipment is a contact scanner known as a Coordinate Measuring Machine (CMM), not a laser scanner. As I understand it, to scan with a laser scanner, the spray must be applied no matter how industrial-level accurate scanners are, so the quality of scan data is no different from that of the test group intraoral scanner. By reflecting the dimensions of each block measured by CMM in the design data (mentioned as 3D design in the manuscript), the data was transformed into the same size as the real milled. And then, this modified data was used as reference data. I want to check whether I understood well. In addition, no matter how accurate measured values are reflected as design data, it is thought that there might still be errors. How did the authors try to overcome this part?

Response: Thank you for your valuable comment. It is clearly correct what you understood as mentioned above. We wanted to make the reference dataset which was not affected by the scanning-aid materials. Of course, there should be inevitable error because it was not obtained by direct scanning. However, we try to reduce the error as possible by reflecting many measured data in the designing software.

Tables 1 and 2 do not contain information on statistical results. In the discussion section, the authors mentioned trueness, but more analysis on precision results should follow. Although there was no statistical difference, the ScanCure group showed high precision, which should be added to the manuscript. The first sentence of the second paragraph in the discussion section says '~ VITA group had lowest RMS values,' but it seems to be the opposite. In the middle of the third paragraph, 'A hardware design of the products was sufficient to reproduce the results,' it is unclear whether the products refer to sprays/liquid, intraoral scanner, or specimen, so it is difficult to understand the intention of the authors.

Response: Thank you for your valuable comment. Fig 4 and 5 are box plots which include data in Table 1 and 2 respectively. 

As confirmed in Fig 5 and Table 2, the IP group showed the highest precision among the three groups with all the reference models although there was no statistical difference. The following sentence was revised in the discussion section.

Line 221: “Unlike trueness, there were no statistically significant differences in the RMS values of precision even though the IP group showed the highest precision among the three groups with all the reference models.” 

The following sentence was revised to correct our mistake.

Line 204: “Furthermore, the VITA group had the largest RMS values among the three groups with all the reference models, with a statistically significant difference.”

The products what we meant were scanning-aid materials. The following sentence was revised to clarify our intention. 

Line 225: “However, in each group, the hardware design of the scanning-aid materials was sufficient to reproduce the results of the scanned images.”

---

## [Decision Letter · Decision Letter 1]

14 Apr 2022

Effect of Scanning-Aid Agents on the Scanning Accuracy in Specially Designed Metallic Models: A Laboratory Study

PONE-D-21-40935R1

Dear Dr. Lim,

We’re pleased to inform you that your manuscript has been judged scientifically suitable for publication and will be formally accepted for publication once it meets all outstanding technical requirements.

Kind regards,

Antonio Riveiro Rodríguez, PhD

Academic Editor

PLOS ONE

Additional Editor Comments (optional):

Please, prior to publication, correct the captions of Figs. 4 & 5 as indicated by reviewer 1

Reviewers' comments:

Reviewer's Responses to Questions

**Comments to the Author**

1. If the authors have adequately addressed your comments raised in a previous round of review and you feel that this manuscript is now acceptable for publication, you may indicate that here to bypass the “Comments to the Author” section, enter your conflict of interest statement in the “Confidential to Editor” section, and submit your "Accept" recommendation.

Reviewer #1: All comments have been addressed

Reviewer #2: All comments have been addressed

Reviewer #3: All comments have been addressed

2. Is the manuscript technically sound, and do the data support the conclusions?

Reviewer #1: Yes

Reviewer #2: Yes

Reviewer #3: Yes

3. Has the statistical analysis been performed appropriately and rigorously? 

Reviewer #1: Yes

Reviewer #2: Yes

Reviewer #3: Yes

4. Have the authors made all data underlying the findings in their manuscript fully available?

Reviewer #1: Yes

Reviewer #2: Yes

Reviewer #3: No

5. Is the manuscript presented in an intelligible fashion and written in standard English?

Reviewer #1: Yes

Reviewer #2: Yes

Reviewer #3: Yes

6. Review Comments to the Author

Reviewer #1: (No Response)

Reviewer #2: Dear authors, thank you for your opinion and activity. I accept that your arguments are correct and in accordance with the opinion of the reviewer.

Reviewer #3: The authors have amended and modified the text adequately with my suggestions for improvement and, to the best of my understanding, also proposed by the other referees. The part that still needs additional correction is that the figure legends of Figures 4 and 5 are identical. Please state that Fig. 4 indicates trueness, and Fig. 5 describes precision. I strongly recommend that this manuscript be published in PLOS ONE.

7. PLOS authors have the option to publish the peer review history of their article (what does this mean?). If published, this will include your full peer review and any attached files.

Reviewer #1: No

Reviewer #2: **Yes: **Professor (Assoc.) Tihomir Dovramadjiev PhD Eng.

Reviewer #3: No

---

## [Editor Report · Acceptance letter]

27 Apr 2022

PONE-D-21-40935R1 

Effect of Scanning-Aid Agents on the Scanning Accuracy in Specially Designed Metallic Models: A Laboratory Study 

Dear Dr. Lim:

I'm pleased to inform you that your manuscript has been deemed suitable for publication in PLOS ONE. Congratulations! Your manuscript is now with our production department. 

Kind regards, 

on behalf of

Dr. Antonio Riveiro Rodríguez 

Academic Editor

PLOS ONE